# Complementary Roles of GCN5 and PCAF in Foxp3+ T-Regulatory Cells

**DOI:** 10.3390/cancers11040554

**Published:** 2019-04-18

**Authors:** Yujie Liu, Chunrong Bao, Liqing Wang, Rongxiang Han, Ulf H. Beier, Tatiana Akimova, Philip A. Cole, Sharon Y. R. Dent, Wayne W. Hancock

**Affiliations:** 1Transplant Immunology, Department of Pathology & Laboratory Medicine & Biesecker Center for Study of Pediatric Liver Diseases, Children’s Hospital of Philadelphia, and Perelman School of Medicine, University of Pennsylvania, Philadelphia, PA 19104, USA; yujieliu2010@gmail.com (Y.L.); bcron@126.com (C.B.); wangl@email.chop.edu (L.W.); hanr@email.chop.edu (R.H.); Tatiana.Akimova@pennmedicine.upenn.edu (T.A.); 2Division of Nephrology, Department of Pediatrics, Children’s Hospital of Philadelphia, and Perelman School of Medicine, University of Pennsylvania, Philadelphia, PA 19104, USA; beieru@email.chop.edu; 3Department of Biological Chemistry and Molecular Pharmacology, Brigham and Women’s Hospital and Harvard Medical School, Boston, MA 02115, USA; pacole@bwh.harvard.edu; 4Department of Epigenetics and Molecular Carcinogenesis, Center for Cancer Epigenetics, University of Texas MD Anderson Cancer Center Science Park, Smithville, TX 78957, USA; sroth@mdanderson.org

**Keywords:** acetyltransferase, Treg cells, epigenetics, transplantation, autoimmunity, anti-tumor immunity, acetylation, chromatin

## Abstract

Functions of the GCN5-related N-acetyltransferase (GNAT) family of histone/protein acetyltransferases (HATs) in Foxp3+ T-regulatory (Treg) cells are unexplored, despite the general importance of these enzymes in cell biology. We now show that two prototypical GNAT family members, GCN5 (general control nonrepressed-protein 5, lysine acetyltransferase (KAT)2a) and p300/CBP-associated factor (p300/CBP-associated factor (PCAF), Kat2b) contribute to Treg functions through partially distinct and partially overlapping mechanisms. Deletion of Gcn5 or PCAF did not affect Treg development or suppressive function in vitro, but did affect inducible Treg (iTreg) development, and in vivo, abrogated Treg-dependent allograft survival. Contrasting effects were seen upon targeting of each HAT in all T cells; mice lacking GCN5 showed prolonged allograft survival, suggesting this HAT might be a target for epigenetic therapy in allograft recipients, whereas transplants in mice lacking PCAF underwent acute allograft rejection. PCAF deletion also enhanced anti-tumor immunity in immunocompetent mice. Dual deletion of GCN5 and PCAF led to decreased Treg stability and numbers in peripheral lymphoid tissues, and mice succumbed to severe autoimmunity by 3–4 weeks of life. These data indicate that HATs of the GNAT family have contributions to Treg function that cannot be replaced by the functions of previously characterized Treg HATs (CBP, p300, and Tip60), and may be useful targets in immuno-oncology.

## 1. Introduction

The development of T-regulatory (Treg) cells that play essential roles in immune tolerance and homeostasis requires both Foxp3 expression and establishment of a specific pattern of CpG hypomethylation [1,2]. Each process has a distinct role in establishing Treg-specific gene expression. Foxp3 represses expression of several key molecules, such as IL-2, IFN-γ and Zap70, whereas Treg cell-specific CpG hypomethylation enhances Ikzf2 and Ikzf4 expression [3,4]. There are two types of Tregs: natural Treg (nTreg) that develop in the thymus, and induced Treg (iTreg) that develop in the periphery from naïve CD4+ T cells upon exposure to TGF-β [5]. A sustained high level of Foxp3 expression is required to maintain both Treg populations. Complementary in vitro and in vivo studies show Foxp3 is regulated by multiple post-translational modifications, including acetylation, ubiquitination, and phosphorylation that affect Foxp3 DNA binding ability and protein stability, and regulate Treg stability, proliferation, and suppressive function [6,7,8,9,10,11].

Members of two of the three main histone/protein acetyltransferase (HAT) families, CBP/p300, GNAT (GCN5/PCAF) and MYST [12], are known to be critical for Treg development. Thus, mice with conditional deletion of both CBP/Kat3a and p300/Kat3b in their Tregs develop severe autoimmunity and die in early life [13], similarly to Scurfy mice that have a loss of function deletion in *Foxp3* [14], while deletion of either CBP or p300 results in only a modest decrease in Treg suppressive function. In addition, mice with conditional deletion of Tip60/Kat5 in their Tregs develop lethal autoimmunity in early life [11,15]. These data illustrate the critical roles that HATs play in controlling Treg development. However, little is known of the role of GNAT family members in Treg biology.

The current study investigated the functions of two GNAT enzymes, general control non-derepressible 5 (GCN5, Kat2a) and p300/CBP-associated factor (PCAF, Kat2b). PCAF and GCN5 are highly homologous proteins, sharing ~73% amino acid sequence identity [16], but corresponding knockout mice have distinct phenotypes. GCN5−/− mice die before birth, whereas PCAF−/− mice have normal lifespans [17,18,19]. The finding that GCN5 mRNA is detectable earlier than PCAF mRNA during embryogenesis may partially explain this difference [19]. GCN5 and PCAF are involved in the regulation of diverse activities, including cell cycle progression, apoptosis, proliferation, innate antiviral immunity, and adipogenesis, and can function via HAT-dependent, HAT-independent or non-transcriptional mechanisms [20,21,22,23,24]. However, there are no data regarding their functions in Foxp3+ Treg cells. The accompanying studies show that deletion of either HAT had modest and partially overlapping but also some distinct, effects on Treg cells, whereas their dual deletion led to severe autoimmunity and death by 3–4 weeks of age. In addition, PCAF was unexpectedly found to regulate iTreg production and Treg stability during TCR stimulation. As a consequence, targeting PCAF was shown to decrease tumor volume and enhance anti-tumor immunity in a Treg-dependent manner and without provoking obvious host autoimmunity. Thus, our studies highlight the roles of GNAT HAT family members in controlling Treg development and function.

## 2. Results

Results are presented in the sequence of single deletion of GCN5, single deletion of PCAF and then their dual deletion. Floxed GCN5 was conditionally deleted in Tregs by mating with Foxp3^cre^ mice, or in all T cells by mating with CD4^cre^ mice, whereas PCAF studies used mice with global gene deletion (PCAF−/−), and studies of dual-targeted mice were focused on Treg cells (Foxp3^cre^Gcn5^fl/fl^/PCAF−/− mice).

### 2.1. No Effects of GCN5 Deletion on Tregs In Vitro but Inhibitory Effects on Treg Function In Vivo

Conditional deletion of GCN5 in the Tregs of GCN5^flf^Foxp3^YFP-cre^ mice (Appendix A) had no significant effect on T cell numbers (Figure 1A) or their baseline level of immune activation (Figure 1B). The proportions of CD4+Foxp3+ Treg cells of the GCN5 conditional KO mice were comparable to their littermates (Figure 1C), including over several months of age (Appendix A), and their suppressive ability (Figure 1D) and stability upon activation in vitro (Figure 1E) were normal. Beyond the thymus, iTregs can be generated in vitro by TCR-stimulation of conventional T-effector (Teff) cells in the presence of TGF-β [5]. GCN5 deletion in CD4+CD25− Teff cells did not affect PCAF expression level compared to WT T cells (Figure 1F), but when cultured for 3 days using standard conditions that promote iTreg development [6], GCN5 deletion had no significant effect on iTreg production in vitro (Figure 1G, Appendix A). Thus, conditional GCN5 deletion had negligible effects on nTreg development, stability or function, or on the development of iTreg cells.

We used 3 models to assess the effects of GCN5 deletion in Tregs in vivo. First, we adoptively transferred CD90.2+CD4YFP+ nTregs isolated to >98% purity by cell-sorting from Foxp3^cre^Gcn5^fl/fl^ or control Foxp3^cre^ mice, along with CD90.1+CD4+CD25− conventional T cells, into B6/Rag1−/− mice. Analysis at two weeks post-transfer showed comparable CD4+Foxp3+ Treg proportions between the two groups (Figure 1H), suggesting comparable stability of the Treg phenotype despite lack of GCN5. Second, we used a model that assessed Treg suppression of homeostatic T cell proliferation [6]. We adoptively transferred CD90.1+CD4+CD25− conventional T cells, alone or with CD90.2+CD4+YFP+ Treg from Foxp3^YFPcre^ or Foxp3^cre^Gcn5^fl/fl^ mice, into B6/Rag1−/− mice and compared conventional T cell numbers at 1-week post-transfer. No significant differences were found between Foxp3^cre^Gcn5^fl/fl^ and control groups (Figure 1I), indicating comparable Treg suppressive function despite lack of GCN5. Third, we used a Treg-dependent model of allograft tolerance in which recipients receive CD154 mAb plus 5 × 10^6^ donor splenocytes, i.v., at the time of cardiac engraftment [6]. Whereas WT mice accepted their grafts long-term (>120 days), allografts in mice whose Tregs lacked GCN5 were subject to acute rejection (Figure 1J). These data indicate that a biologically significant contribution of GCN5 to Treg function can be demonstrated in transplant models, whereas GCN5-deficient Tregs can appear normal in shorter term or less stringent models. 

### 2.2. GCN5 Deletion Had No Effect on Overall T Cell Development but Impaired Teff Cell Functions

GCN5 deletion had no significant effect on T cell development (Appendix A) or immune activation (Appendix A) in CD4^cre^GCN5^fl/fl^ mice, but impaired proliferation (Appendix A) and decreased production of the NFκB-dependent cytokine, IL-2 (Appendix A), upon activation in vitro when compared with cells from WT mice. Western blotting showed that GCN5 promotes acetylation of p65/NFκB (at lysine 310) and histone-H3 in 293 cells and Teff cells, and ChIP studies showed recruitment of GCN5 to the IL-2 promoter in WT Teff cells. GCN5 deletion also decreased Teff cell functions in vivo, as indicated by the prolongation of allograft survival in CD4^cre^GCN5^fl/fl^ versus WT mice (Appendix A, *p* < 0.01). Hence, while conditional GCN5 deletion using CD4cre occurs in both Foxp3+ Treg and conventional Teff cell populations, the net effect in Teff cells was greater as evidenced by the inhibitory effects on IL-2 production and the decreased tempo of allograft rejection.

### 2.3. PCAF Helps Maintain Treg Stability Under TCR Stimulation

To assess the role of PCAF in Tregs, we used PCAF−/− mice that are known to develop normally [18,19]. PCAF deletion in CD4+ T cells did not alter the expression of GCN5 protein (Figure 2A). Compared to corresponding young WT mice (4–6 weeks of age), PCAF−/− CD4+ and CD8+ T cells showed normal thymic development and were present in comparable proportions in peripheral lymphoid tissues (Figure 2B). However, when compared at 6 months of age, CD4+ and CD8+ T lymphocytes from PCAF−/− mice showed mild increases in cell activation, especially in mesenteric lymph nodes, as seen by increases in CD44^hi^CD62L^lo^ and CD4+CD69+ cells (Figure 2C). The proportions of CD4+Foxp3+ cells were comparable between littermate and PCAF−/− mice in thymus and peripheral lymphoid tissues at 4–6 weeks of age (Figure 2D), and also at 6 months (data not shown). Expression of GITR, CD25 and CTLA4 cells were comparable between PCAF−/− and control mice, as was the suppressive function of PCAF−/− Tregs in vitro (Figure 2E). To assess PCAF Treg stability, CD4+CD25^hi^ Tregs were sorted from WT and PCAF−/− mice and stimulated for 24 h with CD3/28 mAb-coated beads. Flow cytometric analysis showed that PCAF−/− Treg had significantly lower proportions of CD4+Foxp3+ cells (Figure 2F) and increased apoptosis (Figure 2G); cumulative data are shown in Appendix A. Hence, while its actions are subtle, PCAF helps protect Tregs from undergoing apoptosis upon TCR stimulation.

### 2.4. PCAF Is Important for iTreg Production

Given an activated CD4+ T cell population in mesenteric lymph nodes of PCAF−/− mice (Figure 2C), we assessed the contribution of PCAF to iTreg development. Naïve CD4^+^CD25^−^CD62L^high^CD44^low^ T cells were sorted from WT and PCAF−/− mice and incubated with IL-2, TGF-β and CD3/CD28 mAbs for 3 days. Compared to WT cells, CD4+ T cells from PCAF−/− mice showed less iTreg development, with significantly lower proportions of CD4+Foxp3+ cells (Figure 3A, Appendix A) and Foxp3 mRNA (Figure 3B), indicating that PCAF is important for Treg peripheral conversion in vitro.

To determine whether PCAF is required for iTreg production in vivo, naïve WT or PCAF−/− conventional T cells were adoptively transferred to Rag1−/− mice, and spleens were harvested 3 weeks later. The percentage (Figure 3C) and absolute number (Figure 3D) of CD4+Foxp3+ Treg cells generated were reduced in the case of PCAF−/− versus WT cells. Both IL-2 and TGF- β cytokines are required for iTreg development [25,26,27,28]. Beginning with IL-2, we assessed whether PCAF deletion affects IL-2 production by stimulating CD4+ T cells from PCAF−/− or control mice overnight with CD3/CD28 mAb-coated beads. Compared to WT CD4+ T cells, PCAF−/− CD4+ T cells had less IL-2 protein production (Figure 3E,F). However, PCAF deletion did not globally diminish CD4+ T cell responses to TCR stimulation, since PCAF−/− CD4+ had higher IFN-γ levels than WT CD4+ T cells (Figure 3G). We also analyzed the effects of PCAF deletion on Th1 cell differentiation. Naïve CD4+CD25−CD62L^hi^CD44^lo^ T cells sorted from PCAF−/− or WT mice were cultured under Th1 polarizing conditions for 4 days. Subsequent intracellular staining showed increased IFN-γ production in PCAF−/− versus WT CD4+ T cells (Appendix A). These data indicate that PCAF promotes IL-2 but represses IFN-γ production. Chromatin immunoprecipitation showed that PCAF deletion in CD4^+^ T cells had less acetylated histone-3 binding at the *il-2* promoter (Figure 3H). To assess whether reduced IL-2 production could cause impaired iTreg production, various amounts of IL-2 were added during iTreg production in vitro. Exogenous IL-2 increased CD4^+^Foxp3^+^ iTreg production using both WT and PCAF^−/−^ cells, with high doses of IL-2 (50 U/mL) rescuing PCAF^−/−^ CD4^+^Foxp3^+^ iTreg formation to that of WT controls (Figure 3I). We next evaluated effects of PCAF deletion on TGF-β signaling. The TCR-stimulated Smad signaling pathway is important for iTreg production, and previous studies showed that PCAF associates with Smad3 and promotes TGF-β/Smad induced transcriptional responses [29]. Naïve CD4+ cells from WT and PCAF−/− were cultured under iTreg conversion conditions, and cell lysates were analyzed by Western blotting for levels of phosphorylated Smad3 (S423/425) and total Smad3. We found that compared to WT cells, naïve CD4^+^ T cells from PCAF^−/−^ cells had normal Smad3 expression but reduced levels of phosphorylated Smad3 (Figure 3J). Hence, these findings suggest that PCAF can regulate iTreg production by affecting both IL-2 production and TGF-β-induced Smad3 phosphorylation.

### 2.5. PCAF Targeting in WT Mice Impairs Inhibits Treg Function In Vitro and In Vivo

We further explored functions of PCAF by undertaking studies with WT Treg cells. Endogenous PCAF co-localized with Foxp3 in the nuclei of murine CD4+CD25+ Tregs, and in 293T cells transfected with Foxp3 and PCAF expression vectors (Figure 4A). Immunoprecipitation of PCAF on 293T cells led to co-precipitation of Foxp3 (Figure 4B), and PCAF expression of PCAF promoted Foxp3 acetylation, as seen by immunoprecipitation with anti-acetyl-lysine Ab and immunoblotting with anti-Foxp3 mAb (Figure 4C). Hence, PCAF can associate with, and acetylate, Foxp3. To test if PCAF inhibition affected Treg function, we undertook Treg suppression assays in the presence of a potent (IC_50_ 0.3 µM) and selective (~200-fold) PCAF inhibitor, H3-CoA-20-Tat peptide [30,31,32]. Compared to control peptide, H3-CoA-20-Tat decreased Treg function in vitro (Figure 4D). We then used a transplant model in which Tregs suppress host alloresponses and promote long-term graft survival [6]. Immunodeficient (B6/Rag1−/−) C57BL/6 mice were engrafted with BALB/c hearts, adoptively transferred with 1 × 10^6^ conventional T cells alone or with 0.5 × 10^6^ Tregs, and treated with Tat peptides for 14 days. Treatment with H3-CoA-20-Tat led to acute rejection by 3 weeks post- transplant, whereas the control Tat peptide had no effect on long-term allograft survival for at least >120 days (Figure 4E). Hence, PCAF can acetylate Foxp3, and its pharmacologic targeting can impair Treg function Treg function, including in the context of Treg-dependent allograft survival, whereas the control Tat peptide had no effect on long-term allograft survival for at least >120 days (Figure 4E). Hence, PCAF can acetylate Foxp3, and its pharmacologic targeting can impair Treg function, including in the context of Treg-dependent allograft survival.

### 2.6. GCN5 and PCAF Deletion in nTregs Causes Lethal Autoimmunity

To gauge the importance of GCN5 and PCAF in Foxp3+ Treg development and function, we crossed Foxp3^YFP-cre^GCN5^flfl^ mice with PCAF−/− mice to develop what we term hereafter as double knockout (DKO) mice (Figure 5A). As with our report of mice with Treg-specific deletion of CBP and p300 [13], DKO mice were runted (Figure 5B) and experienced the rapid onset of autoimmunity and death by 3–4 weeks of age (Figure 5C,D, and Table 1). 

Compared to WT mice, thymii of DKO mice were markedly atrophic, whereas their lymph nodes and spleens showed significant cellular expansion (Figure 5E and Appendix A). CD4+ T cells were highly activated compared to littermates, as indicated by significantly increased proportions of CD44^hi^CD62L^lo^, CD4+CD69+, and CD4+CD25+ cells (Figure 5F and Appendix A), and by their significantly increased proliferation rate (CD4+Ki67+%) (Figure 5G and Appendix A). Additionally, compared to littermates, CD4+ T cells from DKO had higher expression of IFN-γ but lower levels of IL-2 levels in lymph nodes and spleen, and higher IL-4 and IL-17 expression in lymph node cells (Figure 5H and Appendix A). These data indicate that the dual deletion of GCN5 and PCAF in Tregs promotes conventional T cell activation and proliferation and leads to rapidly lethal autoimmunity.

Upon direct analysis of Treg cells in DKO mice, the proportions of CD4+Foxp3+ cells in peripheral lymphoid tissues were significantly lower than in littermates (Figure 6A and Appendix A), whereas thymic levels were comparable. This difference in peripheral Treg numbers could reflect impaired nTreg proliferation but GCN5/PCAF deletion in Tregs did not impair Treg cell division (Figure 6B). To assess Treg stability in vitro, highly purified CD4+Foxp3+ Tregs (isolated by sorting CD4+YFP+ cells to >98% purity) were cultured for 3 days in the presence of CD3/28 mAbs and IL-2. Under these activating conditions, DKO Tregs down-regulated Foxp3 and had less Foxp3/cell (determined by mean fluorescence intensity, MFI), compared to control Tregs (Figure 6C). At the same time, levels of GATA3, a transcription factor required for the maintenance of high levels of Foxp3 expression in Tregs [33], were markedly lower in DKO Tregs versus WT controls (Figure 6D).

To assess effects of dual HAT deletion on Treg survival in vivo, Tregs were isolated by cell sorting from Foxp3^YFP-cre^ and DKO mice (CD90.2+ background) and adoptively transferred, i.v., along with CD90.1+CD4+CD25− conventional T cells, into B6/Rag1−/− mice. Analysis after 4 weeks post-transfer showed that the percentage and absolute numbers of CD4+Foxp3+Tregs were significantly lower in B6/Rag1−/− mice receiving DKO versus WT Tregs (Figure 6E). Collectively, these data indicate that GCN5 and PCAF are important for maintaining Treg stability in vivo and in vitro. 

Since deletion of either CBP/p300 [13] or GCN5/PCAF (Figure 6C) affected Treg stability and caused early lethality, we tested whether all 4 HATs are required for high Foxp3 expression. We found that the levels of Foxp3 protein and acetylated Foxp3 were increased when CBP, p300, GCN5 and PCAF were co-expressed together compared to levels associated with CBP and/or p300, or GCN5 and/or PCAF (Figure 6F). Hence, all four HATs likely contribute to Foxp3 expression, though deletion of either p300 or CBP had only modest effects on Treg biology compared to the effects of their dual deletion [10,13], and likewise with individual deletion of GCN5 or PCAF. These findings indicate that that the members of the same HAT family, e.g., p300/CBP and GNAT/PCAF, can at least partially compensate for the deletion of each other, but that the GNAT and p300/CBP families each have essential and non-redundant roles in Treg biology.

### 2.7. PCAF and Anti-Tumor Immunity

Foxp3+ Treg cells are known to suppress protective host immune responses to a wide variety of solid tumors, but their therapeutic targeting is largely restricted to their transient depletion or “secondary” modulation, e.g., using anti-CTLA-4 mAb [34,35]. We have previously shown that certain lung cancer lines, such as TC1 lung adenocarcinomas [36], grow as syngeneic tumors in WT mice and that anti-tumor immunity exerted by host CD8 T cells in these mice is restrained by Foxp3+ Treg cells; e.g., conditional deletion of p300 or CBP in Foxp3+ Treg cells, or use of a pharmacologic inhibitor directed again p300 and CBP, restores anti-tumor immunity and limits tumor growth [10]. Based on our current findings, we hypothesized that GNAT targeting might have similar effects, and assessed the effects of PCAF targeting on lung tumor growth in syngeneic mice. Growth of TC1 lung adenocarcinomas was impaired in PCAF−/− mice compared to WT mice (Figure 7A), in conjunction with decreased infiltration by CD4+Foxp3+ Treg cells (Figure 7B) but increased infiltration by host CD8 T cells (Figure 7C). Hence targeting PCAF can decrease tumor volume and increase antitumor immunity in syngeneic mice. 

## 3. Discussion

HATs are grouped into two types, A and B, based on their intracellular localization and substrate specificity. The paralogs Gcn5 and PCAF are both members of the A-type HAT class, since they are located primarily in the nucleus and promote acetylation of nucleosomal histones [37,38]. Both enzymes have an ~160-residue HAT domain and a C-terminal bromodomain and have a strong preference, among histones, for acetylation of H3K14 [38]. Both HATs are ubiquitously expressed, with GCN5 functioning as part of the SAGA and ATAC complexes [38,39], whereas PCAF can be autoacetylated or acetylated by p300, and both HATs can acetylate a variety of non-histone, as well as histone, proteins [40]. Gcn5-deleted mice die during embryogenesis, with failure to develop dorsal mesodermal structures, including the neural tube [18,19]. By contrast, PCAF−/− mice appear largely normal [17], though problems with learning [41] and neurodegeneration [42,43] are reported. Neither HAT has, to our knowledge, been studied in the context of Foxp3+ Treg cells.

Deletion of GCN5 or PCAF had negligible effects on nTreg development or in vitro suppressive functions, but in each case, deletion decreased iTreg development when cells were cultured under polarizing conditions. The effects on iTreg development were greater in the case of PCAF deletion, and this event also decreased Treg stability. The effects of deletion of each gene were most obvious when the respective mice were challenged with cardiac allografts, since recipients with GCN5- of PCAF-deficient Tregs were unable to suppress acute allograft rejection. Likewise, WT allograft recipients treated with a PCAF inhibitor also acutely rejected their cardiac transplants. 

The finding of normal Treg function in vitro versus impaired Treg function in vivo may reflect several factors. First, the in vitro Treg suppression assay has significant limitations. It is primarily dependent upon cell–cell contact and is best used as a screening tool. Second, the lack of GCN5 or PCAF may be compensated, through as yet undefined mechanisms, by the presence of the other GNAT family member, whereas loss of both, as discussed below, is catastrophic to Treg function. Third, the importance of GCN5 and PCAF in Tregs may differ. Thus, allograft recipients with global T cell deletion of GCN5 showed prolonged allograft survival, indicating the inhibitory effects of GCN5 loss on Teff cell functions are greater than that of loss in Treg cells. By contrast, the acute rejection developing in conjunction with targeting of PCAF in WT allograft recipients indicates that the brunt of the effect of PCAF inhibition fell on Tregs rather than Teff cells. The translational significance of these findings is that pharmacologic targeting of GCN5 but not PCAF may of therapeutic significance in allograft recipients, though appropriately selective and HAT-specific non-peptidic inhibitors are still unknown.

Dual deletion of GCN5 and PCAF had far more profound effects than targeting either HAT alone, with death from systemic autoimmunity by 3–4 weeks of life. This outcome is analogous to that seen when we deleted both CBP and p300 within Foxp3+ Tregs, deletion of either HAT alone had only modest effects on Treg suppressive function, whereas dual deletion, like in the current study, led to death from overwhelming autoimmunity by 3–4 weeks of life [13]. Despite apparently normal thymic Treg production, dual deletion of GCN5 and PCAF in Tregs led to reduced Treg stability and markedly decreased numbers in peripheral lymphoid tissues, with corresponding widespread activation of Teff cells. While members of both GNAT and CBP/p300 HAT families can associate with, and acetylate, Foxp3, it remains unknown whether specific HATs are required for acetylation of specific lysine residues, as appears to be the case for Tip60, which is essential for acetylation-dependent Foxp3 dimerization [15,44], or if regulation of at least partially differing sets of transcription factors or other proteins are responsible. Just as multiple HDACs play distinct roles in Treg cells [45], it is increasingly apparent that their HAT counterparts also have important and differing functions. 

Unravelling these diverse roles may have important consequences for therapeutic targeting of Tregs in various disease settings ranging from autoimmunity to cancer. Indeed, the first data concerning the effects of PCAF targeting in tumor-bearing mice, shown here, supports this contention. Clearly, the mice concerned lacked PCAF in all immune cells. However, the current tumor data are consistent with our allograft findings. Together, these data strongly support the conclusion that deletion of PCAF particularly affects Tregs versus non-Treg cells, since the net effect of PCAF deletion was to promote effector T cell responses in allografted or tumor-bearing mice. Further insights as to the efficacy of PCAF targeting in tumor models will likely arise as specific inhibitors are identified. Progress in that direction is occurring, with small molecules such as L-Moses being reported; this compound has a Ki of 47 nM against PCAF and 600 nM against GCN5, and has no significant activity against a panel of 48 other bromodomains [46]. In future studies, we plan to test this and additional recently developed compounds with GNAT activity in syngeneic tumor models.

## 4. Materials and Methods

### 4.1. Mice and Cardiac Allografting

We purchased BALB/c and C57BL/6 mice (The Jackson Laboratory, Bar Harbor, ME, USA), and CD4^cre^ mice [47], Foxp3^YFP-cre^ mice [48], PCAF−/− [49], and Gcn5^fl/fl^ mice [49] were described previously. Mice were housed under specific-pathogen-free conditions and studied using protocols approved by the Institutional Animal Care and Use Committee of the Children’s Hospital of Philadelphia (#2011-7-746 and #2016-6-561). 

### 4.2. Antibodies, Flow Cytometry and Cell Sorting

We purchased Pacific blue-CD4, APC-Foxp3, PE-Cy7-CD62L, APC-CD44, APC-CD25, APC-GATA3, PE-Cy5.5-Ki67, APC-IFN-γ, PE-IL-2 and PE-IL-4 monoclonal antibodies (mAbs) from BD Pharmingen; mAb to mouse Foxp3 (FJK-16s, eBioscience, San Diego, CA, USA); and rabbit antibodies to β-actin, Flag Tag and acetyl-lysine (Cell Signaling, Danvers, MA, USA). Rabbit anti-acetylated-Foxp3 (acetylated-K31) antibody was produced at Covance (Princeton, NJ, USA). We used a Cyan ADP Color flow cytometer (Beckman Coulter, Pasadena, CA, USA), and data were analyzed by FlowJo 8 software (TreeStar, Ashland, OR, USA). CD4+YFP+ Tregs were sorted, based on YFP expression, with a FACS-Aria fluorescent cell sorter (BD Biosciences, Billerica, MA, USA) at the UPenn Flow Cytometry Core Facility. 

### 4.3. Plasmids

PCAF and p300 expression vectors were provided by Xiao-Jiao Yang (McGill University, Montreal, QC, Canada). CBP and GCN5 expression vectors were purchased from Addgene. Foxp3-MinR1 was described previously [50].

### 4.4. Treg Assays

CD4+CD25− T-effector (Teff) and CD4+CD25+ Treg cells were isolated from WT and PCAF−/− mice using magnetic beads [10]. CD4^+^YFP^+^ Treg or CD4^+^YFP^−^ Teff were sorted from Foxp3^YFP-cre^ and GCN5^flfl^/Foxp3^cre^ based on YFP expression. Pacific blue- or CFSE-labeled Teff cells (4 × 10^5^) were stimulated with CD3 mAb (5 μg/mL) in the presence of 5 × 10^5^ irradiated, syngeneic, T-cell depleted splenocytes, and varying ratios of Tregs. Teff cell proliferation was determined by flow cytometric analysis of Pacific blue or CFSE dilution 72 h later [10]. To assess Treg stability in vivo, we adoptively transferred, i.v., 0.1 million FACS-sorted high-purity Tregs (CD90.2+CD4+YFP+) along with 0.5 million Teff cells (CD90.1+CD4+CD25−) into B6/Rag1−/− mice. Spleens were harvested after 2–4 weeks for flow cytometric analysis. In Treg conversion studies, FACS-sorted naïve CD4+CD25−CD44^lo^CD62L^hi^ cells (0.2 × 10^6^) were adoptively transferred to B6/Rag1−/− mice, and spleens were harvested at 3 weeks for flow cytometric analysis. 

### 4.5. Western Blotting

Protein was extracted from cell lysates, quantified, and separated by SDS-PAGE, transferred to polyvinylidene fluoride membranes and subjected to immunoblotting using the indicated Abs.

### 4.6. Quantitative PCR (qPCR) and ChIP Assays

Total RNA was extracted using a RNeasy Kit (Qiagen, Valencia, CA, USA), and cDNA was synthesized with TaqMan reverse transcription reagents (N808-0234, Applied Biosystems, Wilmington, DE, USA) following the manufacturer’s instructions. Real-time PCR was performed using TaqMan Universal PCR Master Mix and primers from Applied Biosystems. ChIP assays were performed with an EZ-Magna CHIP A/G ChIP Kit (Upstate, 17-408). DNA-chromatin complexes were isolated from 2.5 × 10^6^ CD4+ T cells, and genomic DNA precipitated using anti-AcH3 Ab (Upstate, 06-599) or control IgG. Genomic DNA was probed by qPCR for IL-2 promoter [51].

### 4.7. Co-Localization Studies

293T cells transfected with Foxp3 and PCAF constructs, and cytospins of Treg-enriched mononuclear cells, were stained with antibodies to Foxp3 and PCAF and assessed by confocal microscopy.

### 4.8. Homeostatic Proliferation

CD90.1+CD4+CD25− T-cells (1 × 10^6^) plus CD4+YFP+ Tregs (0.5 × 10^6^), sorted from Foxp3^YFP-cre^ or GCN5^flfl^Foxp3^YFP-cre^ mice (CD90.2+), were adoptively transferred to B6/Rag1−/− mice. After 7 days, spleens were isolated and total CD90.1+CD4+ T-cells determined by flow cytometry.

### 4.9. Cardiac Transplant Studies

We performed heterotopic cardiac allografting, as described [6], using WT BALB/c donors and C57BL/6 recipients that were WT or had conditional deletion of Gcn5 within all T cells (CD4^cre^Gcn5^fl/fl^) or just within Foxp3+ T-regulatory (Treg) cells (Foxp3^cre^Gcn5^fl/fl^). Recipients were injected with rapamycin (0.1 mg/kg/day, i.p.) for 14 days postoperatively. Allograft survival was assessed by daily palpation, and rejection was confirmed by histologic evaluation of H&E-stained paraffin sections [6]. In adoptive transfer studies, heterotopic cardiac allografts were performed using BALB/c donors and B6/Rag1−/− recipients (*n* = 5/group). Recipients were adoptively transferred with CD4+CD25− T cells alone (1 × 10^6^), or along with CD4+CD25+ Tregs (5 × 10^5^), and Alzet pumps were implanted subcutaneously to deliver control peptide or H3-CoA-20-Tat infusions.

### 4.10. TC1 Tumor Growth

TC1 cells were derived from mouse lung epithelial cells that were immortalized with HPV-16 E6 and E7 and transformed with the c-Ha-ras oncogene [36]. For tumor studies, WT and PCAF−/− mouse were shaved on their right flanks and each injected s.c. with 1.2 × 10^6^ TC1 tumor cells. Tumor volume was determined by the formula: (3.14 × long axis × short axis × short axis)/6.

### 4.11. Statistical Analyses

Data were analyzed using GraphPad Prism and are displayed as mean ± standard deviation or standard error of mean. Measurements between two groups were performed with Student’s *t*-test (* *p* < 0.05, ** *p* < 0.01, *** *p* < 0.001).

## 5. Conclusions

The current data are of both basic and translational significance. From a basic science perspective, our studies provide the first insights as to the functions of key HATs of the GNAT family in Foxp3+ Treg cells. Unexpectedly, targeting of GCN5 prolonged allograft survival, whereas targeting of PCAF shortened allograft survival, revealing markedly contrasting functions of these enzymes in Tregs despite their close structural homology. From a translational perspective, PCAF-selective targeting may be of therapeutic significance in the field of immune-oncology, given the effects of PCAF deletion on promoting antitumor immunity and curtailing tumor growth in syngeneic mice. Lastly, despite the relatively modest effects of deletion of either gene on Foxp3+ Treg development and function in vitro, their dual deletion resulted in the rapid onset of lethal autoimmunity, indicating the inability of other GNAT family members, as well as the various non-GNAT HATs, to compensate for their loss.

## Figures and Tables

**Figure 1 cancers-11-00554-f001:**
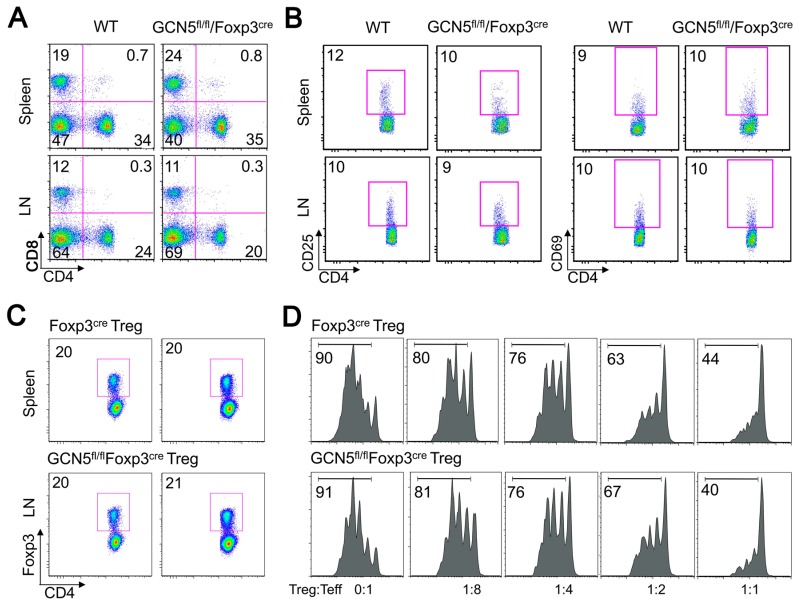
Minor effects of general control nonrepressed-protein 5 (GCN5) deletion on T-regulatory cells (Tregs) in vitro but inhibitory effects on Treg function in vivo. (**A**) Percentages of CD4+ and CD8+ T cell population in peripheral lymph node and spleens of Foxp3^YFP-cre^ and Foxp3^YFPcre^GCN5^flfl^ mice. (**B**) Comparison of percentages of CD4+CD25+, CD4+CD69+ cells in peripheral lymph nodes and spleens of Foxp3^YFPcre^ and Foxp3^YFPcre^GCN5^flfl^ mice. (**C**) Comparison of percentages of CD4+Foxp3+ Tregs in lymph nodes, mesenteric lymph nodes, and spleens of Foxp3^YFPcre^ and Foxp3^YFPcre^GCN5^flfl^ mice. (**D**) CD4+YFP+ Tregs were sorted from Foxp3^YFPcre^ and Foxp3^YFPcre^GCN5^flfl^ mice, incubated with Pacific Blue Cell-Tracer labeled Foxp3^cre^CD4+YFP− T-effector (Teff cells), irradiated APC, and CD3 mAb for 3 days. The percentage of proliferating Teff cells is shown in each panel. (**E**) CD4+yellow fluorescent protein (YFP)+ cells (1 × 10^5^) were sorted from Foxp3^YFPcre^ or Foxp3^YFPcre^GCN5^flfl^ mice, stimulated in vitro with plate-bound CD3/CD28 mAbs for 24 hr. The percentages of CD4+Foxp3+ cells are shown. (**F**) CD4+ T cells isolated from wild-type (WT) and CD4^cre^GCN5^flfl^ mice and analyzed by western blot; β-actin was used as a loading control. (**G**) CD4+CD25− Teffs were isolated from WT and CD4^cre^GCN5^flfl^ mice and stimulated with CD3/28 mAb-coated beads, TGF-β and IL-2 for 3 days; the percentages of CD4+Foxp3+ iTregs are shown; though lack of GCN5 led to decreased iTreg development, this reduction was not statistically significant (Appendix A). (**H**) CD90.1+CD4+CD25− Teff cells (1 × 10^5^) were adoptive transferred, along with CD90.2+CD4+YFP+ Tregs (1 × 10^5^) from Foxp3^YFPcre^ or Foxp3^YFPcre^GCN5^flfl^ mice, into B6/Rag1−/− mice and percentages of CD90.2+CD4+YFP+ Tregs were assessed at 2 weeks post-transfer. (**I**) Homeostatic proliferation model in which Rag1−/− mice were adoptively transferred with CD90.1+CD4+CD25− Teff cells (5 × 10^5^) alone, or in conjunction with CD90.2+CD4+YFP+ Tregs (1 × 10^5^) isolated by cell sorting from Foxp3^YFPcre^ or Foxp3^YFPcre^GCN5^flfl^ mice; the absolute numbers of Teff cells present in pooled lymph nodes and spleen at 7 days post-transfer are shown (*p* < 0.01 for Teff alone versus mice receiving WT or GCN5−/− Tregs). (**J**). Kaplan–Meier survival curves showing acute rejection of MHC-mismatched cardiac allografts (BALB/c to C57BL/6) in Foxp3^YFPcre^GCN5^flfl^ mice whose Tregs lack GCN5 versus WT recipients (*n* = 4, *p* < 0.01); all mice received 10 mg/kg/d of rapamycin for 14 days from the time of engraftment. Data are representative of 3 independent experiments.

**Figure 2 cancers-11-00554-f002:**
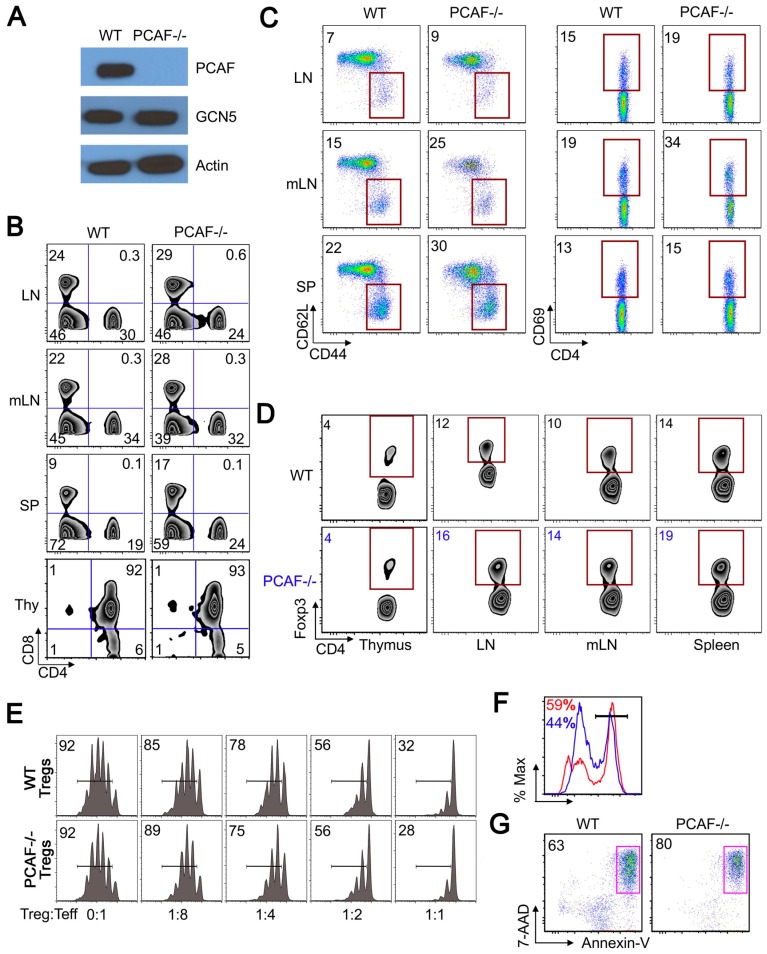
p300/CBP-associated factor (PCAF) is important for inducible Treg (iTreg) production. (**A**) CD4+ T cells were isolated from WT and PCAF−/− mice and analyzed by Western blotting, with β-actin as a loading control. (**B**,**C**) Data are from 6-month-old mice. (**B**) Comparison of CD4+ and CD8+ T cell percentages in peripheral lymph nodes, mesenteric lymph nodes, spleens, and thymii of WT and PCAF−/− mice. (**C**) Comparison of percentages of CD44^hi^CD62L^lo^ and CD4+CD69+ cells in WT and PCAF−/− mice. (**D**) Percentages of CD4+Foxp3+ Tregs in thymii, peripheral lymph nodes, mesenteric lymph nodes, and spleens of WT and PCAF−/− mice. (**E**) CD4+CD25+ cells were isolated from WT and PCAF−/− mice and incubated with CFSE-labeled WT CD4+CD25− Teff cells, irradiated APC, and CD3 mAb for 3 days; the percentages of proliferating Teff cells are shown. (**F**,**G**) CD4+CD25^hi^ cells were isolated from WT and PCAF−/− mice and stimulated with CD3/CD28 mAb-coated beads for 24 h. (**F**) The percentages of Foxp3+ cells are shown, with cells gated on CD4+; red indicates WT Tregs and blue indicates PCAF−/− Tregs, *p* < 0.05 as seen in Appendix A. (**G**) The percentages of apoptotic 7-AAD+ Annexin V+ are shown, and cells were gated on CD4+Foxp3+ staining; *p* = 0.02 as seen in Appendix A. Data are representative of 3 independent experiments.

**Figure 3 cancers-11-00554-f003:**
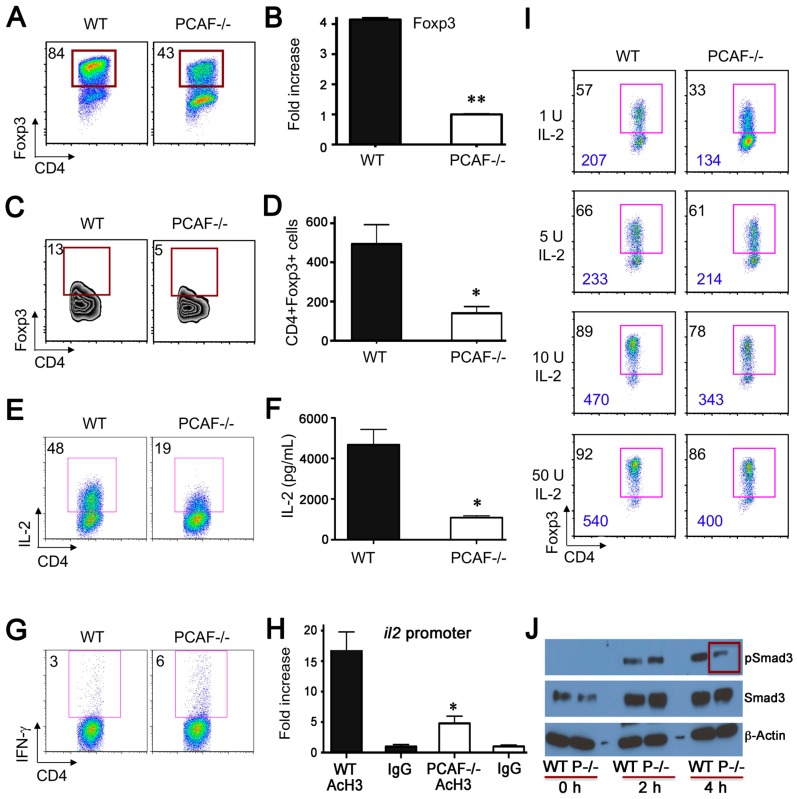
PCAF targeting in WT mice impairs inhibits Treg function in vitro and in vivo. (**A**) Sorted naïve CD4+CD44^lo^CD62L^hi^ CD25− T cells were stimulated with CD3/CD28 mAb-coated beads, IL-2 and TGFβ for 3 days; percentages of CD4+Foxp3+ Tregs are shown (* *p* < 0.05, Appendix A). (**B**) qPCR analysis of Foxp3 mRNA expression using sorted CD4+CD44^lo^CD62L^hi^CD25− naïve T cells stimulated with CD3/CD28 mAb-coated beads, IL-2 and TGF- β for 1 day; ** *p* < 0.01. (**C**,**D**) Sorted CD4+CD25−CD44^lo^CD62L^hi^ naïve T cells (2 × 10^5^) from WT or PCAF/- were adoptively transferred i.v. to B6/Rag1−/− mice for 3 weeks. Spleens were harvested and (**C**) CD4+Foxp3+ percentages and (**D**) absolute numbers determined; * *p* < 0.05. (**E**–**G**) CD4+ T cells were isolated from WT and PCAF−/− mice and stimulated with CD3/CD28 mAb-coated beads for 24 h, followed by (**E**) intracellular staining of IL-2, (**F**) measurement of supernatant IL-2 protein by ELISA (* *p* < 0.05), and (**G**) intracellular staining of IFN-γ. (**H**) CD4+ T cells were isolated from WT and PCAF−/− mice and analyzed by ChIP-qPCR assay for the levels of acetyl-H3 at the *il2* promoter (* *p* < 0.05). (**I**) Percentages of CD4+Foxp3+ cells generated when varying amounts of IL-2 were added to the iTreg conversion conditions described in the legend for panel (**A**); MFI level is shown in blue. (**J**) Sorted CD4+CD44^lo^CD62L^hi^CD25− naïve T cells were stimulated with CD3/CD28 mAb-coated beads, IL-2 and TGF-β for the periods indicated, and cell lysates analyzed by western blotting. Data are representative of 3 independent experiments.

**Figure 4 cancers-11-00554-f004:**
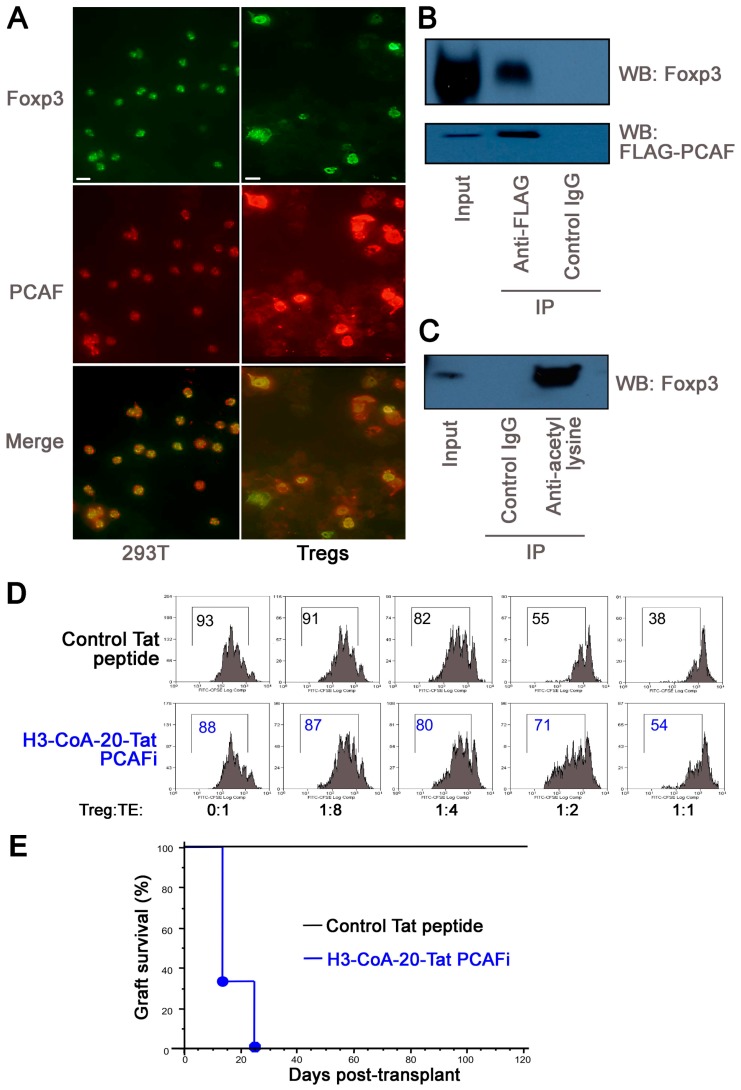
PCAF helps maintain Treg stability under TCR stimulation. (**A**) Colocalization of Foxp3 and PCAF in nuclei of transfected 293T cells, and colocalization of endogenous Foxp3 and PCAF in nuclei of Tregs; Foxp3 (green), PCAF (red) and cells with co-localization appear in yellow. (**B**,**C**) 293T cells were transfected with Foxp3 and PCAF-Flag expression vectors for 48 h. Scale bars = 10 microns. (**B**) Cell lysates were immunoprecipitated with anti-Flag (Ac-K) or anti-rabbit-IgG Abs followed by western blotting with anti-Foxp3 Abs. (**C**) Cell lysates were also immunoprecipitated with anti-Ac-K or anti-rabbit-IgG Abs followed by Western blotting with anti-Foxp3 mAb. (**D**) CD4^+^CD25+ Tregs were isolated from C57BL/6 mice and incubated with CFSE-labeled CD4+CD25− Teff cells, irradiated APC, and CD3 mAb in the presence of control Tat peptide or H3-20-CoA-Tat (10 µM) for 3 days. Percentages of proliferating Teff cells are shown. (**E**) Kaplan–Meier plots of BALB/c cardiac allograft survival in B6/Rag1−/− mice (5 mice/group) that were adoptively transferred with WT Teffs (1 × 10^6^) and Tregs (5 × 10^5^) and treated with control Tat peptide or H3-20-CoA-Tat (10 µM) via Alzet pumps from the time of engraftment. Data are representative of 2 independent experiments.

**Figure 5 cancers-11-00554-f005:**
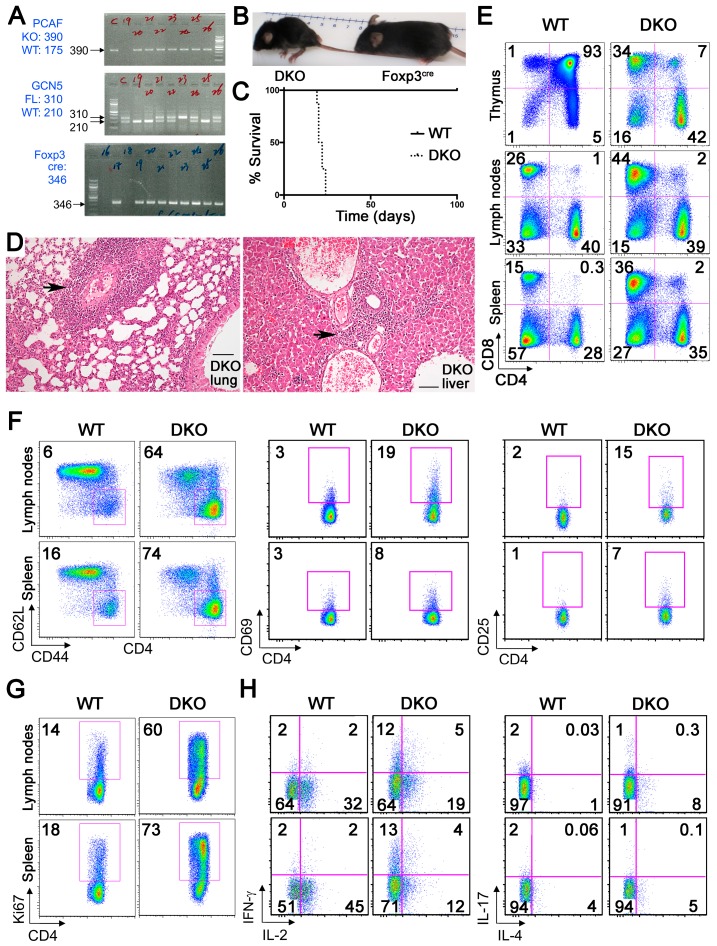
GCN5 and PCAF deletion in Foxp3+ Treg cells causes lethal autoimmunity. (**A**) Development of double knockout (DKO) mice. (**B**) Appearance at 3 weeks of age. (**C**) Kaplan–Meier survival curves. (**D**) Representative histology of lungs and livers in DKO mice at 3 weeks of age; arrows indicate abnormal mononuclear cell infiltrates. Scale bars = 20 microns. (**E**) Percentages of CD4+, CD8+ T cells in thymii, lymph nodes and spleens of WT versus DKO mice. (**F**) Percentages of CD44^hi^CD62L^lo^, CD4+CD69+ and CD4+CD25+ cells in lymph nodes and spleens of WT versus DKO mice (**G**) Percentages of CD4+Ki67+ cells in lymph nodes and spleens of WT versus DKO mice. (**H**) Intracellular cytokine staining (IL-2, IFN-γ, IL-4 and IL-17) by CD4+ Teff cells from WT versus DKO mice. Cells were gated on CD4 expression, following isolation of single cells from lymph nodes and spleens, and stimulation with PMA/ionomycin for 4 h. Data are representative of 3 independent experiments.

**Figure 6 cancers-11-00554-f006:**
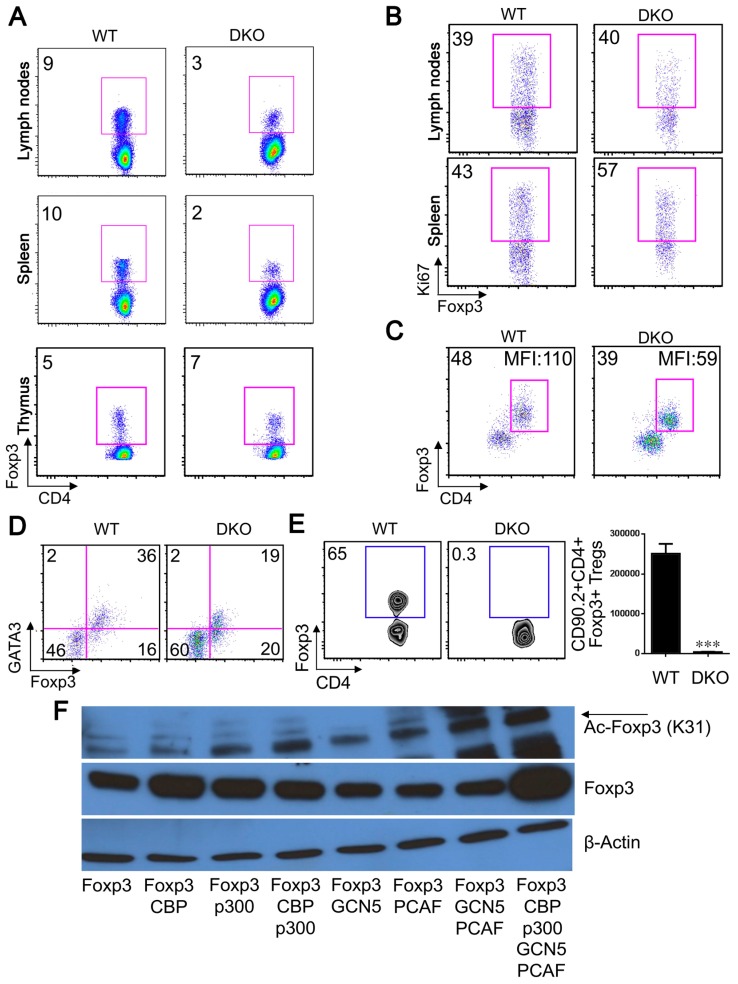
GCN5 and PCAF are critical for Treg stability. (**A**) Percentages of CD4^+^Foxp3^+^ level in lymph nodes, spleens, and thymii of DKO and littermate control mice at 3 weeks of age. (**B**) Percentages of Foxp3+Ki67+ cells in WT and DKO mice; cells were gated on CD4+Foxp3+ cells. In panels (**C**,**D**), CD4+YFP^+^ Tregs (1 × 10^5^) were sorted from Foxp3^YFP^ or DKO mice, stimulated in vitro with plate-bound CD3/CD28 mAbs plus IL-2 for 3 days. (**C**) CD4+Foxp3+ percentage and mean fluorescent intensity (MFI) shown in each panel. (**D**) Percentages of GATA3+ in CD4+Foxp3+ cells are shown. (**E**) Percentage and absolute numbers of CD90.2+CD4+Foxp3+ cells at 4 weeks after i.v. adoptive transfer of CD4+YFP+ Treg cells (1 × 10^5^), isolated by cell sorting from Foxp3^cre^ or DKO mice, into B6/Rag1−/− host mice; each sorted Treg population was co-transferred with CD4+CD25−CD90.1+ Teff cells (5 × 10^5^) and *** *p* < 0.001. (**F**) 293T cells were transfected with Foxp3 alone or plus CBP, p300, GCN5, PCAF; after 48 h, cell lysates were subjected to Western blotting as indicated, and β-actin was used as a loading control. Data are representative of 3 independent experiments.

**Figure 7 cancers-11-00554-f007:**
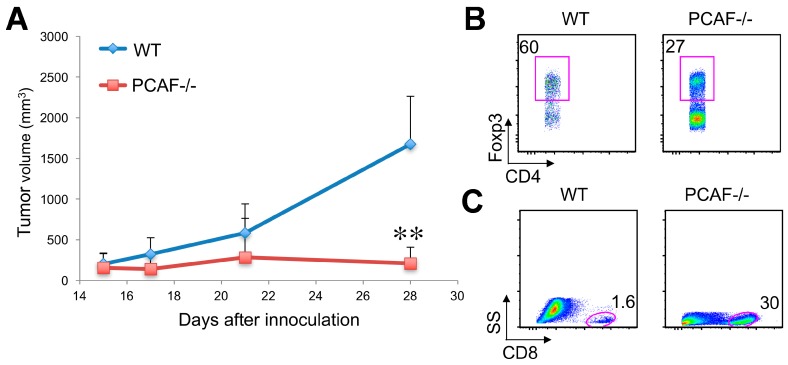
PCAF targeting inhibits tumor growth. (**A**) TC1 tumor growth in WT and PCAF−/− C57BL/6 mice (tumor volumes shown as mean ± SD, ** *p* < 0.01 by Mann–Whitney *U* test). Additional panels show flow cytometric analyses of the frequency of (**B**) CD4+Foxp3+ Tregs and (**C**) CD8+ T cells population within TC1 tumors. Data are representative of 2 separate experiments, with 10 mice/group.

**Table 1 cancers-11-00554-t001:** Histopathology of tissues from male Foxp3^cre^GCN5^flfl^PCAF−/− mice (3 wks of age) ^1^.

Tissue	Histology
Lungs	Lungs show moderate to marked perivascular and peribronchiolar lymphocytic infiltrates primarily centered around bronchovascular bundles and also extends to distal airspaces with widening of intra-alveolar septae.
Liver	Hepatic architecture is intact with mild to moderately dense lymphocytic infiltration involving the majority of large and small portal tracts. Inflammation is also present within the lobules and there is diffuse extramedullary hematopoiesis, characterized by scattered collections of erythroid precursors and occasional megakaryocytes, within sinusoids.
Spleen	Splenic parenchyma has accentuation and expansion of peri-arteriolar lymphoid sheaths. Collections of mature lymphocytes and foci of extramedullary hematopoiesis, characterized by numerous megakaryocytes and erythroid precursors, are present in the red pulp.
LN	Lymph nodes are cellular, but there is effacement of nodal architecture with loss of distinct follicles and germinal center formation is not seen.
Thymus	Thymic tissue is markedly diminished in volume and the corticomedullary junction is indistinct. Cortical thymocytes appear mature.
Skin	A diffuse mononuclear inflammatory infiltrate is predominantly composed of small mature lymphocytes within the superficial dermis, with focal exocytosis.
Brain	Brain features are normal without significant leptomeningeal or parenchymal inflammation.
Pancreas	Variable, mild lymphocytic inflammation is present focal around islets of Langerhans, and with infiltration into adjacent peri-pancreatic adipose tissues.
Heart	Normal cardiac myocytes and vasculature.
Kidney	Normal histology.
Gut	Small bowel mucosa has scattered collections of mononuclear cells in the lamina propria, without infiltration into surface epithelium, and villous architecture is intact

^1^ H&E-stained paraffin sections from 4 mice were compared with corresponding littermate and WT controls.

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
