# Peer review of "Complementary Roles of GCN5 and PCAF in Foxp3+ T-Regulatory Cells"

_cancers, 2019, doi:10.3390/cancers11040554_

Reviewer 1 Report

HATs is Histone acetyltransferase or Histone/protein acetyltransferase?

GCN5(fl/fl) is miss-labeling.

There is no statistical labelling in the Figures 1, 2, &5.

1. It's hard to convince us according to n = 1 data. Authors indicated 3 independent experiments. Why didn't authors show bar chart with marked with significance?

2. In the description of Fig 1B, the authors did not present any stimulation process within this experiment. How can the authors say "no significantly immune activation" in here?

3. In Figure 1C, the dot density of each cluster was not consistent in 4 sub-figures. What is the total events in each sub-figures?

In Fig. 3  *,** represent what?? And what is comparison???

In Fig. 6  *** represent what?? And what is comparison???

In Fig. 7  ** represents what?? And what is comparison???

When the authors tried to the role of PCAF in in vitro and mouse model,  experimental examinations should run on the CpG hypomethylation to double check the epigenetic occurrence.  

In the translational aspect, the authors speculated GCN5 and PCAF could be applied in the hope of cancer target therapy, is this GNAT HATs found in autoimmune disease or any connection within autoimmune such as SLE or MS caused cancer.

Author Response

Responses to reviewers of cancers-436448

Complementary Roles of GCN5 and PCAF in Foxp3+ T-Regulatory Cells (Liu et al)

Our responses are shown in blue.

Reviewer 1

HATs is Histone acetyltransferase or Histone/protein acetyltransferase?

As clearly defined in the Abstract and, as is the convention, again upon first use in the text, we introduce the abbreviation of HATs for histone/protein acetyltransferases. The latter term reflects the ability of these enzymes to promote acetylation of >2000 non-histone proteins, just as in some papers the point is made by referring to them as lysine acetyltransferases (KATs). There was no problem with our use of histone/protein acetyltransferases (HATs) in our recent paper in Nature Medicine (see Abstract of that paper in the footnote) or others we have published on this topic.[1]

GCN5(fl/fl) is miss-labeling.

Tough to know what is misleading or mislabeled here; the floxed GCN5 mice are clearly described in the Methods section and appropriately referenced (reference 49). The originator of these floxed mice and senior author of reference 49, Dr. Sharon Dent, is a co-author of the current paper and a leading investigator in the field of epigenetics.

There is no statistical labelling in the Figures 1, 2, &5.

This point is addressed below.

1. It's hard to convince us according to n = 1 data. Authors indicated 3 independent experiments. Why didn't authors show bar chart with marked with significance?

It is standard with flow cytometry to show actual flow plots to demonstrate the quality of the data being discussed.  However, where we did find significant differences or when the point was of added importance, cumulative data are now also shown in the Supplement. For example, with regard to Fig. 1G, cumulative data are now shown in a new Fig. S1C and a corresponding comment made in the text as to the lack of significant difference in iTreg development. Additions of other relevant cumulative data are noted below and include new Fig. S7, Fig. S9, Fig. S10 and Fig. S11.

2. In the description of Fig 1B, the authors did not present any stimulation process within this experiment. How can the authors say "no significantly immune activation" in here?

We have clarified our meaning by restating this as “Conditional deletion of GCN5 in the Tregs of GCN5flfFoxp3YFP-cre mice (Figure S1a) had no significant effect on T cell numbers (Figure 1A) or baseline level of immune activation (Figure 1B)”. I.e. We are referring to the existing level of immune activation of cells as they are seen in the mice at baseline; often with our studies, including that in the footnote, targeting of a gene in Tregs decreases their function and leads to T cell activationin vivo.

3. In Figure 1C, the dot density of each cluster was not consistent in 4 subfigures. What is the total events in each sub-figures?

We collected sufficient cell numbers to provide the results; the total events in each sub-figure of Fig. 1C were:

Foxp3cre

GCN5Foxp3cre

Spleen

269376

189117

LN

84257

74918

In Fig. 3 *,** represent what?? And what is comparison???

We now note in the Figure legend that * indicates p£0.05 and ** indicates for data comparing PCAF-/- and WT mice. We have also added cumulative data in support of Figures 2 and 3 (new Fig. S7).

In Fig. 6 *** represent what?? And what is comparison???

We have added in the Figure legend that *** indicates p£0.001, clearly referring to the numbers of Tregs (y-axis) present at 4 weeks post-transfer of WT vs. DKO cells.

In Fig. 7 ** represents what?? And what is comparison???

As already indicated in the Figure legend, this refers to the differing tumor volumes at day 28 in WT vs. PCAF-/- mice.

When the authors tried to the role of PCAF in in vitro and mouse model, experimental examinations should run on the CpG hypomethylation to double check the epigenetic occurrence.

I am not sure quite what the reviewer is referring to here, but our paper is the first analysis ever of the biologic significance of GCN5 and PCAF in Foxp3+ Treg cells.

In the translational aspect, the authors speculated GCN5 and PCAF could be applied in the hope of cancer target therapy, is this GNAT HATs found in autoimmune disease or any connection within autoimmune such as SLE or MS caused cancer.

Again, I am not sure quite what the reviewer is referring to here, but our paper is the first analysis ever of the biologic significance of GCN5 and PCAF in Foxp3+ Treg cells, and as we show herein, while individual deletion does not have profound effects, their dual deletion leads to death from the rapid onset of lethal autoimmunity.

[1]Inhibition of p300 impairs Foxp3⁺ T regulatory cell function and promotes antitumor immunity. Liu Y, Wang L, Predina J, Han R, Beier U, Wang LC, Kapoor V, Bhatti T, Akimova T, Singhal S, Brindle P, Cole P, Albelda S, Hancock WW. Nat Med. 2013 Sep;19(9):1173-7. 

Forkhead box P3 (Foxp3)+ T regulatory (Treg) cells maintain immune homeostasis and limit autoimmunity but can also curtail host immune responses to various types of tumors1,2. Foxp3+ Treg cells are therefore considered promising targets to enhance antitumor immunity, and approaches for their therapeutic modulation are being developed. However, although studies showing that experimentally depleting Foxp3+ Treg cells can enhance antitumor responses provide proof of principle, these studies lack clear translational potential and have various shortcomings. Histone/protein acetyltransferases (HATs)promote chromatin accessibility, gene transcription and the function of multiple transcription factors and nonhistone proteins3,4. We now report that conditional deletion or pharmacologic inhibition of one HAT, p300 (also known as Ep300 or KAT3B), in Foxp3+ Treg cells increased T cell receptor–induced apoptosis in Treg cells, impaired Treg cell suppressive function and peripheral Treg cell induction, and limited tumor growth in immunocompetent but not in immunodeficient mice. Our data thereby demonstrate that p300 is important for Foxp3+ Treg cell function and homeostasis in vivo and in vitro, and identify mechanisms by which appropriate small-molecule inhibitors can diminish Treg cell function without overtly impairing T effector cell responses or inducing autoimmunity. Collectively, these data suggest a new approach for cancer immunotherapy.

Reviewer 2 Report

The article from Yujie Liu presents an interesting and relevant research topic for immuno-oncology and questions of autoimmunity or transplant rejection. Overall, the design of their project is well built.
I would be in favor of publishing this article when the major point and the three minor points will be reviewed.

Major point:
1- The big problem for me is the lack of statistical information. A large majority of the article is based on flow cytometry analyzes. Even if the representative dot plots are of good quality, I think it is essential that the authors present in graphical form the data of the different mice (WT, DKO, GCN5, PCAF) with a statistical analysis and not only representative dot plots.

2- The representative images of the IF in FIG. 4A are of very poor quality. This point needs to be improved. Authors should consider using a more appropriate method for assessing protein colocalisation (confocal microscopy, duolink). It seems to me that this part is not described in the material and method part.

Minor points:
1- The representative images of the WT mice for Figure 5D (lungs and liver histology) are missing.
2- The WB of Figures 6F (Ac-Foxp3) can not be presented in this form.
It needs to be improved.

After major points modification, I think that this article could be publish in Cancer journal.

Author Response

Reviewer 2

Our responses are shown in blue.

The article from Yujie Liu presents an interesting and relevant research topic for immuno-oncology and questions of autoimmunity or transplant rejection. Overall, the design of their project is well built. I would be in favor of publishing this article when the major point and the three minor points will be reviewed.

Major point:

1- The big problem for me is the lack of statistical information. A large majority of the article is based on flow cytometry analyzes. Even if the representative dot plots are of good quality, I think it is essential that the authors present in graphical form the data of the different mice (WT, DKO, GCN5, PCAF) with a statistical analysis and not only representative dot plots.

In response, we have now added key cumulative data in Fig. S1 and Figs. S7-11, encompassing 13 new histograms with statistical analyzes.

2- The representative images of the IF in FIG. 4A are of very poor quality. This point needs to be improved. Authors should consider using a more appropriate method for assessing protein colocalisation (confocal microscopy, duolink). It seems to me that this part is not described in the material and method part.

We were not able to repeat this work given that the respective embryos are now cryo-preserved and we are no longer carrying the actual mice in our colony. However, the contrast of the Figure 4 has been improved.

Minor points:

1- The representative images of the WT mice for Figure 5D (lungs and liver histology) are missing.

In response, arrows have been added to indicate the abnormal infiltrates (if we added histology of normal WT lung and liver, the respective images will become smaller and smaller).

2- The WB of Figures 6F (Ac-Foxp3) can not be presented in this form. It needs to be improved. 

With respect, I disagree. The blot involves simultaneous analysis of the acetylation of a single amino acid of Foxp3 after simultaneous transfection of up to 5 genes of interest. This work is technically complex and there is no significant problem with the Western blot. 

After major points modification, I think that this article could be publish in Cancer journal.

Reviewer 3 Report

In this manuscript, the authors analyzed two members of the GNAT family of histone/protein acetyltransferases (HATs): GCN5 and PCA for their ability to affect inducible Treg (iTreg) development in vitro and in vivo. 

The results presented in this manuscript well supported the hypothesis of the authors, that were able to identify two contrasting functions of these enzymes in Tregs since they observed that the targeting of GCN5 prolonged allograft survival, while on the contrary targeting of PCAF

 shortened allograft survival.

Minor Concerns:

In figure 4A a the authors performed an immunofluorescence assay to demonstrate the co-localization of PCAF  with Foxp3 in the nuclei of murine CD4+CD25+ Tregs, and in 293T cells transfected with Foxp3 and PCAF expression vectors. In this experiment, proper staining of the nuclei should be performed to demonstrate the presence of PCAF and Foxp3 in this cellular compartment.

Author Response

Reviewer 3

Our responses are shown in blue.

In this manuscript, the authors analyzed two members of the GNAT family of histone/protein acetyltransferases (HATs): GCN5 and PCA for their ability to affect inducible Treg (iTreg) development in vitro and in vivo.  The results presented in this manuscript well supported the hypothesis of the authors, that were able to identify two contrasting functions of these enzymes in Tregs since they observed that the targeting of GCN5 prolonged allograft survival, while on the contrary targeting of PCAF shortened allograft survival.

Minor Concerns:

In figure 4A the authors performed an immunofluorescence assay to demonstrate the co-localization of PCAF with Foxp3 in the nuclei of murine CD4+CD25+ Tregs, and in 293T cells transfected with Foxp3 and PCAF expression vectors. In this experiment, proper staining of the nuclei should be performed to demonstrate the presence of PCAF and Foxp3 in this cellular compartment.

We were not able to repeat this work given that the respective embryos are now cryo-preserved and we are no longer carrying the actual mice in our colony. However, the contrast of the Figure 4 has been improved.

Round  2

Reviewer 1 Report

L255-266 (figure legend) should be put underneath Fig. 4.

The ref 1 (L576-578) and ref 40 (L681-682), the format without vol and page needs to be modified.

It is still the same too many results which just presented via flowcytometric gating plot, the authors encourage to show the statistical format as a bar chart with standard error becuase the data is easy to find individual variation of every measurement.   

Author Response

Response to Comments of Reviewer 1 (cancers-436448)

Complementary Roles of GCN5 and PCAF in Foxp3+ T-Regulatory Cells (Liu et al)

Our responses are shown in blue.

Reviewer 1

L255-266 (figure legend) should be put underneath Fig. 4.

This was done.

The ref 1 (L576-578) and ref 40 (L681-682), the format without vol and page needs to be modified.

The references were corrected.

It is still the same too many results which just presented via flowcytometric gating plot, the authors encourage to show the statistical format as a bar chart with standard error becuase the data is easy to find individual variation of every measurement.   

During the previous revision, we added key cumulative data in Fig. S1 and Figs. S7-11, encompassing 13 new histograms with statistical analyzes; these new data were placed in the Supplement and perhaps were not seen by the reviewer.